# Automatic Detection of Dyspnea in Real Human–Robot Interaction Scenarios

**DOI:** 10.3390/s23177590

**Published:** 2023-09-01

**Authors:** Eduardo Alvarado, Nicolás Grágeda, Alejandro Luzanto, Rodrigo Mahu, Jorge Wuth, Laura Mendoza, Richard M. Stern, Néstor Becerra Yoma

**Affiliations:** 1Speech Processing and Transmission Laboratory, Electrical Engineering Department, University of Chile, Santiago 8370451, Chile; eduardo.alvarado@ug.uchile.cl (E.A.); nicolas.grageda.u@gmail.com (N.G.); alejandro.luzanto@ug.uchile.cl (A.L.); rmahus@gmail.com (R.M.); jwuths@gmail.com (J.W.); 2Hospital Clínico Universidad de Chile, Santiago 8380420, Chile; lmendoza@hcuch.cl; 3Clínica Alemana, Santiago 7630000, Chile; 4Department of Electrical and Computer Engineering, Carnegie Mellon University, Pittsburgh, PA 15213, USA; rms@cs.cmu.edu

**Keywords:** respiratory-distress evaluation, human–robot interaction, deep learning, beamforming methods

## Abstract

A respiratory distress estimation technique for telephony previously proposed by the authors is adapted and evaluated in real static and dynamic HRI scenarios. The system is evaluated with a telephone dataset re-recorded using the robotic platform designed and implemented for this study. In addition, the original telephone training data are modified using an environmental model that incorporates natural robot-generated and external noise sources and reverberant effects using room impulse responses (RIRs). The results indicate that the average accuracy and AUC are just 0.4% less than those obtained with matched training/testing conditions with simulated data. Quite surprisingly, there is not much difference in accuracy and AUC between static and dynamic HRI conditions. Moreover, the beamforming methods delay-and-sum and MVDR lead to average improvement in accuracy and AUC equal to 8% and 2%, respectively, when applied to training and testing data. Regarding the complementarity of time-dependent and time-independent features, the combination of both types of classifiers provides the best joint accuracy and AUC score.

## 1. Introduction

Human–robot communication will play a key role in countless applications in the next decades. Currently, multiple aspects of manufacturing have been improved by the inclusion of robots to improve and optimize their processes [1]. Most robots used are considered only as tools, because they are programmed for specific tasks that do not require much deliberation [2]. On the other hand, social robots are designed to interact with people to achieve common goals. These types of robots are much more relevant in areas such as education and healthcare [3]. 

To emulate human–human communication successfully, a social robot should be able to characterize the users’ profiles either physically, cognitively, or socially [4]. By doing so, the robot could adapt its response based on the users’ behaviors or needs. Physical characterization of a person may require often invasive methods such as those needed to measure blood pressure or lung capacity. Other less invasive options correspond to the use of wearable devices that allow measurements of sleep, movement, neurological activity, or heart rate in a fast and more comfortable fashion for the user [5]. However, wearable sensors are not always accurate or robust, so improving their performance remains a major challenge [6,7]. In this context, the use of voice emerges as a potentially valuable alternative for physical profiling. Speech includes linguistic and paralinguistic information (e.g., prosody) that are especially useful in several applications [8] but that also convey information to detect respiratory problems [9].

### 1.1. Voice-Based Estimation of Respiratory Distress

Chronic respiratory diseases (CRDs) are important to global health systems, particularly regarding bronchial asthma and chronic obstructive pulmonary disease (COPD). It is estimated that there are 262 million people affected by bronchial asthma and more than 200 million people affected by COPD, which causes almost 3 million deaths per year and accounts for 6% of all deaths [10]. Although there is no cure for COPD, its treatment can help improve patients’ quality of life by providing better symptom control [11]. 

Radiography is an inexpensive and commonly used method for the detection and monitoring of respiratory diseases. Computed tomography can also provide visual and quantitative information on disease severity [12]. Spirometry can also detect lung disease, but the result is not easily interpretable [13]. Sound analysis of the respiratory system, including lung, cough, breath, voice, and heart sounds, is used by health professionals to identify respiratory diseases such as asthma, bronchitis, Bordetella Pertussis infection and SARS-CoV-2 [14]. However, all these methods require patients to attend an interview clinic or health center to undergo the relevant tests, which may limit accessibility for patients depending on their ages or locations. 

The significant increase in demand for health services in recent years has led to the development of remote health monitoring tools [15]. Machine learning (ML) has been used to prevent and manage COPD by collecting and integrating large-scale medical data for precision medicine [16]. The COVID-19 pandemic has further driven research on artificial intelligence (AI)-based solutions, such as automatic detection of SARS-CoV-2 [17,18,19,20], mainly on smartphones [21]. These systems enable remote disease monitoring and reduce the need for patients to visit medical centers [22,23,24]. In addition, remote monitoring has been considered to be useful for people with severe chronic illness who have been discharged from the hospital [25].

Automation can also improve scalability by removing the requirement for an expert to assess each individual, and multiple automated healthcare applications have emerged in recent years. An example is the automated detection of COVID-19 or other respiratory diseases using X-ray or CT images of the lungs as input [26]. Alternatively, some studies have focused on the automatic detection of SARS-CoV-2 by analyzing the sounds produced by coughing, vocalization, and forced breathing [27,28]. 

According to [29], voice-based respiratory disease detection can be standardized and can reduce the variability or bias between different physicians administering questionnaires. One example is the Dyspnea Assessment Questionnaire, which can be applied by physicians easily but is not suitable for ordinary people, particularly the elderly who may struggle with understanding and answering the questions. Additionally, the patients´ responses may be affected by their mood or habituation to the disease [30,31,32]. 

Coughing is a prevalent symptom of colds and respiratory illnesses, accounting for up to 38% of respiratory inquiries [33], and can provide valuable information for ML-based models. However, asking the user to repeat the coughing events can affect the naturalness of the symptom and may cause discomfort. Studies have also demonstrated that coughing is not as effective as reading a text or using a sustained vowel for classifying respiratory diseases such as COVID-19 [34].

As mentioned above, automated voice-based detection of respiratory diseases has mainly focused on COVID-19 [35]. However, some studies have also included diseases such as asthma, bronchitis, and pertussis [36]. Surprisingly, the severity of symptoms has not been addressed exhaustively in the literature even though it is an important factor for patient diagnosis and monitoring. In [37], a method for classifying patients with COPD into different severity grades using FEV1 (forced expiratory volume in one second) as the gold classification metric is proposed. Two classification scenarios were considered separately: discerning between non-COPD/COPD individuals and discriminating between mild and moderate COPD according to the FEV1 scale. However, the study did not evaluate the classification between all severity grades simultaneously. In [38], a system to evaluate dyspnea with the mMRC scale on the phone by employing deep learning was described. The method is based on modeling the spontaneous behavior of subjects while pronouncing controlled phonetizations. Remote monitoring of respiratory diseases by means of audio analysis is an important topic in current research. The early identification of dyspnea can be helpful to identify the diseases that cause it (e.g., respiratory or cardiac diseases, etc.) and to allow their timely management. Moreover, monitoring dyspnea is very important to detect acute diseases (such as pneumonia due to COVID-19 and other pneumonias) and thus to determine whether it is necessary to hospitalize. Audio-based remote monitoring can also be useful to follow up the patients´ evolution and to manage chronic conditions.

Mel-frequency cepstral coefficients (MFCCs) and Mel-frequency spectrograms, which have been widely used in automatic speech recognition (ASR), have also been employed for voice-based COVID-19 and breathing distress detection [39,40]. Additionally, the speech signal dynamics can be represented using the first and second derivatives of previous static features [41]. In studies such as [42], other characteristics such as pitch, jitter, and shimmer were also suggested for COVID-19 detection.

The databases used for training machine learning models, both public and private, such as COSWARA [43], DICOVA [44], and COUGHVID [45], share some common characteristics in the recorded audio recordings. These similarities include the use of sustained vowels, breathing sounds, sentence reading, or forced coughing.

In [35,46,47,48,49], the ML architecture and hyperparameters were optimized to obtain deep features with convolutional neural networks (CNNsƒ) layers to address the problems of COVID-19 or respiratory distress detection. These deep features can be concatenated and input to a neural network-based classifier trained on an end-to-end basis to combine the parameters. Staged training can also be employed where classification modules are trained independently with each set of features, and then they are combined to deliver the final system decision. By doing so, each classifier is optimized individually, and fusion methods can be employed, which is not possible with a single neural network architecture. In [49], for example, the final decision is made by feeding the outputs of the classification modules (softmax) into an SVM. The classifier outputs are subjected to the majority vote rule to deliver the final decision in [50]. In [51], the final classification decision is obtained by weighting the output probabilities.

Surprisingly, the optimization of the complementarity that can be provided by different types of phonetizations has not been addressed exhaustively. In some cases, as in [34], the VGG19 CNN architecture was employed to find the vocalization that could provide the highest accuracy in post-COVID-19 patient identification. In other studies, as in [36], the features extracted from the phonetizations are concatenated and input to a neural network that is expected to learn how to combine them.

Interestingly, the problem of optimizing the complementarity observed in different types of vocalizations has not been addressed elsewhere in great detail. In [34], the CNN VGG19 architecture was used to determine the phonetization that could offer the greatest accuracy in identifying patients with COVID-19. In [36], features obtained from several phonetizations are input into a neural network that is designed to learn how to integrate them. The combination of three types of phonetizations is exhaustively explored in [38] to estimate respiratory distress with telephone speech.

### 1.2. Human–Robot Interaction

Although the confinements caused by the pandemic may have come to an end, problems in healthcare centers persist, such as lack of supplies, shortage of professionals, and the growth of vulnerable populations. This is an ideal environment for the introduction of social robots into this world, either attending or collaborating with the health personnel to diagnose or treat the patients. Social robots are already becoming widespread in the healthcare field because of these problems [52] but usually performing administrative tasks or care for children, the elderly, and people with reduced mobility [53,54]. Surprisingly, there has been no significant use of social robots to estimate dyspnea, especially considering the progress that has been made independently in respiratory assessment and the importance of user profiling in HRI in the last years. A clear application could be in a health center where a robot can interview patients without the need of a health professional. Increase in the measured mMRC score can trigger immediate actions to treat the patients. However, it is worth highlighting that the applicability of user profiling goes beyond the health sector. Consider, for example, the use of robots and human beings working together on a common task in a commercial or defense application. Robots may be employed to estimate the degree of respiratory distress of the persons with whom they interact to optimize the task assignments, for example. We know that physical tiredness also leads to some degree of respiratory difficulties. While prompting the users to pronounce controlled vocalizations may not be feasible in some scenarios, this research is an important step toward evaluating the degree of respiratory distress in human–robot collaboration contexts.

As noted above, several studies claim that the voice is a valuable source of information for the detection of respiratory disorders. It is important to bear in mind that the voice can be susceptible to external conditions, such as ambient noise (e.g., cocktail party effect [55]), reverberation, and speaker movement. Therefore, it is crucial to consider these variables when performing voice analysis in the evaluation of respiratory disorders. Several speech enhancement techniques have emerged to address these problems and obtain a cleaner target speech [56]. When there is an array of microphones, it is possible to apply beamforming or spatial filtering schemes generating a higher gain in the desired direction [57]. Classical beamforming mechanisms such as delay-and-sum (D&S) [58] and minimum variance distortionless response (MVDR) [59] are widely used in the literature.

With the progress of AI, deep learning-based beamforming schemes have achieved important results [60]. Studies propose the use of neural networks to estimate time–frequency masks with which covariance matrices can be estimated to be employed in beamforming methods [61,62]. On the other hand, deep learning-based single-channel masking schemes such as TasNets have also achieved significant improvements in speech separation and enhancements by avoiding phase synchronization when reconstructing the signals [63,64]. These single-channel-based techniques have led to methods that perform speech enhancement on each channel individually and then apply beamforming, seeking to reduce the effect of artifacts generated by the masks [65,66,67]. Other approaches have been proposed to carry out multichannel speech enhancement without beamforming, either through the use of autoencoders [68,69], graph neural networks [70], or fully convolutional networks [71].

Generally, other evaluations of systems that perform speech separation or speech enhancements have been performed using simulated databases [72], which makes it more difficult to assess the extent to which these approaches can be applied in the real world. Moreover, it is even more difficult to find studies that assess the performance of the models under real dynamic conditions such as in mobile HRI scenarios. Some exceptions can be found, for example, in [73,74,75], where the performance of speech separation or enhancement systems are evaluated from the ASR point of view in real dynamic conditions. 

In this paper, an automatic dyspnea detection system is adapted for real static and dynamic HRI scenarios. The method presented here enables the monitoring of respiratory distress by employing the modified Medical Research Council (mMRC) scale that classifies dyspnea in five levels, from zero (healthy) to four (very severe). Surprisingly, this is the first study that addresses the problem of respiratory difficulty in HRI despite the relevance of user profiling in the field of robotics. While it is also possible that the manifestation of dyspnea does not depend on the illness or condition that cause it, discriminating dyspnea depending on its cause is beyond the scope of this paper. The aim of the technology presented here was to detect dyspnea in HRI scenarios independently of the underlying cause.

The database that was employed to train the system was initially recorded over the telephone network [38] and is composed of three controlled vocalizations of the user taking deep breaths and then gasping for air, which were designed to represent the user’s behavior while performing them. The first two phonetizations, which are /ae-ae/ and /sa-sa/, provide pertinent information regarding the amount of air that the individuals in question inhale. The third one corresponds to counting from 1 to 30 as quickly as possible to evaluate the subjects’ spontaneous behavior, which requires effort to accomplish. The 1-to-30 counting is inspired by the Roth Test [76]. The target is to provoke involuntary breathing, voice pauses, coughing, tone variation, and other symptoms that could indicate the severity of dyspnea [38]. The database was re-recorded on a real robotic platform under static and dynamic conditions. To train the dyspnea estimation neural networks, an acoustic model of the target room was incorporated into the original data that were convolved with 33 real room impulse responses (RIRs) as in [75], and several environmental noises were added at an SNR between 5 dB and 15 dB. 

As was performed by the system discussed in [38], both time-dependent and time-independent features are extracted from each vocalization to train a separate classifier for each phonetization and type of feature. By doing so, it is possible to reduce the dimensionality of the input vector and take advantage of the complementarity of different vocalizations more explicitly. Time-independent features were classified by multilayer perceptron (MLP) classifiers, while time-dependent features for the /ae-ae/ and /sa-sa/ vocalizations were classified by CNN-based architectures. For the 1-to-30 counting task, a hybrid architecture combining a CNN and long short-term memory (LSTM) neural network was used [38].

In contrast to [77], where results with only simulated acoustic environments were presented, the results reported here were obtained with a real robotic platform that was set up for this study with a PR2 robot. The respiratory distress system was trained on the original telephone data with the same data but after incorporating the acoustic model described above. The D&S and MVDR beamforming schemes were also assessed. The testing subsets included simulated and real HRI conditions with static and dynamic scenarios. In initial experiments using the static and dynamic HRI data, we trained the respiratory distress classifiers using the original telephone data and a model of the target acoustic environment model that was obtained by adding external and/or robot noise and simulating reverberation through the use of estimated RIRs for the task. This system provided an average accuracy and AUC (area under the ROC curve) that are just 0.4% worse than those obtained with matched training and testing conditions using simulated data. The results presented in this paper indicate incorporating the acoustic model into the training subset can lead to increases in accuracy and AUC equal to 13% and 5%, respectively, using real HRI testing data when compared with the original telephone speech training set. Interestingly, the differences between static and dynamic real HRI scenarios in accuracy and AUC of respiratory distress estimation were found to be quite small (less than 1.1%) when the training database was processed to include the acoustic model. The complementarity analysis of time-dependent and time-independent features revealed that the combination of both types of classifiers led to better accuracy than each classifier individually. It is worth mentioning that time-dependent features worked better in the static condition and time-independent parameters in the dynamic one. Regarding the binary dyspnea presence–absence classification, time-dependent features did not help with both static and dynamic HRI scenarios. The best AUC was achieved with the time-independent parameters only. The contributions of the paper are: (a) the first study to address the problem of respiratory distress estimation in real HRI scenarios; (b) the adaptation of a respiratory distress estimation system based on deep leaning to real HRI conditions; (c) the evaluation of acoustic modeling and beamforming techniques in the framework of the problem tackled here; and (d) the evaluation of the complementarity between time-dependent and time-independent features with static and dynamic real HRI scenarios.

## 2. Beamforming

Beamforming technologies (also referred to as spatial filtering methods) are widely adopted to tackle distant speech processing problems. They play a very important potential role in social robotics for both speech-based HRI and audio sources analysis. It is worth highlighting that beamforming technologies can be decomposed into two sub-problems: source localization and the generation of the beamforming to achieve spatial filtering. In this paper, the direction of arrival (DOA) is assumed to be known. Two schemes were evaluated here: D&S and MVDR.

### 2.1. Delay-and-Sum

A microphone array is a collection of multiple microphones that can be combined and processed to achieve spatial filtering with beamforming. This technique can help to reduce noise and reverberation, especially by suppressing non-direct path acoustic signals. In this study, a linear microphone array was employed using the Microsoft Kinect. This device is widely used in HRI applications and features a four-channel linear microphone array (see Figure 1), along with standard RGB and depth cameras. 

Delay-and-sum is a well-known beamforming technique, which involves summing delayed signals to steer the look direction to the direction of arrival (DOA) of sound waves. This produces destructive interference in all directions except for the DOA. The delayed signals are summed to generate the output signal *y*(*t*) as in [78]:(1)$$y\left(t\right)=\sum_{l=1}^{L}x_{l}\left(t-\tau_{l}\right)$$
where $x_{l}\left(t\right)$ denotes the signal samples from microphone l; $\tau_{l}$ is the delay applied to channel *l* with respect to the reference microphone, i.e., microphone 1 in this case; and *L* is the number of channels, i.e., *L* is equal to four here. Delay $\tau_{l}$ is given by [78]:(2)$$\tau_{l}=\frac{d_{l}}{v}\left(sen\left(\phi \right)\right)$$
where $d_{l}$ is the distance between the microphone l and the reference one (see Figure 1) and v is the propagation speed of sound.

### 2.2. MVDR

MVDR is a more advanced technique than D&S and improves beamforming noise suppression by adaptively reducing spatially correlated noise. This is achieved by creating nulls on the interfering signals without affecting the gain in the look direction. If $x_{l}\left(m,i\right)$ represents the ith sample in frame m of channel l, where 1≤m≤M, M is the number of frames in a given utterance, 1≤i≤frameLength, frameLength denotes the frame length in number of samples, and $X_{l}\left(m,\omega \right)$ is obtained by applying the DFT to frame $x_{l}\left(m,i\right)$ and denotes the component at discrete frequency ω in frame m and channel l, where 0≤ω<numFreqBins, $numFreqBins=\frac{DFT\;size}{2}+1$, and DFT size corresponds to the number of samples employed by the DFT. The DFT of the MVDR output at the *mth^h^* frame, $Y\left(m,\omega \right)$, can be estimated as [79]:(3)$$Y\left(m,\omega \right)=w\left(m,\omega \right)\left[\begin{array}{c} X_{1}\left(m,\omega \right)\\ X_{2}\left(m,\omega \right)\\ .\\ .\\ .\\ X_{L}\left(m,\omega \right) \end{array}\right]$$
where the weights are estimated on a frame-by-frame basis as [80]:(4)$$w^{H}\left(m,\omega \right)=\frac{v^{H}\left(m,\omega \right)\sum_{N}^{-1}\left(m,\omega \right)}{v^{H}\left(m,\omega \right)\sum_{N}^{-1}\left(m,\omega \right)v\left(m,\omega \right)}$$

Equation (4) includes the steering vector $v\left(m,\omega \right)={\left[e^{-j\omega \tau_{1}\left(m\right)},e^{-j\omega \tau_{2}\left(m\right)},\ldots ,e^{-j\omega \tau_{L}\left(m\right)}\right]}^{T}$ and the covariance matrix of the noise $\sum_{N}\left(m,\omega \right)=E\left\{N\left(m,\omega \right)N^{H}\left(m,\omega \right)\right\}$, where $E\left\{·\right\}$ denotes the expectation operator and $N\left(m,\omega \right)$ denotes all the frames with only noise.

## 3. Testing Databases Using Static and Dynamic HRI Scenarios

The telephone database is similar to the one used in [38]. It includes patients with COPD, pulmonary fibrosis, and post-COVID-19 patients who were recruited at the Clinical Hospital of University of Chile (HCUCH), as well as healthy volunteers from the Faculty of Physical and Mathematical Sciences (FCFM) at the same university. The study was approved by the scientific ethic committees at the FCFM and HCUCH, and all participants in the database gave informed consent. A pulmonologist from the HCUCH evaluated the severity of their dyspnea using the mMRC scale (the gold standard), which, in turn, was employed as the reference for training the deep learning-based architecture. Participants were instructed to perform the three controlled vocalizations mentioned above without pausing after taking deep breaths. These three phonetizations provide complementary information among them, which, in turn, can lead to improvements when combined.

The database was composed of 100 people, consisting of 66 patients with respiratory problems (43 COPD, 19 pulmonary fibrosis, and 4 post-COVID-19 patients) and 34 healthy individuals. The deep learning models were trained using the mMRC scores as references, with four levels ranging from zero to three. Class Four was underrepresented, with only three individuals, so it was merged with Class Three. Class Zero denotes a healthy person and Class Three corresponds to the most severe level of dyspnea observed in this study. The database included 200 audios per each type of phonetization (2 repetitions per person per type of phonetization), resulting in a total of 600 vocalizations. An automated speech recognition system was used to discriminate the target phonetization from background noise or unwanted audio as in [38]. It is important to note obtaining this kind of clinical data is difficult and time consuming, and it requires an infrastructure that is not easy to achieve. Under these circumstances, one needs to make the best use of the limited number of patients that is available. Demographic and clinical information pertaining to these patients is presented in Table 1.

### 3.1. Robotic Platform and Indoor Environment

To perform the recording of the database, a testbed similar to the one used in [74,75,81] was implemented, where “Jarvis”, a PR2 robot, was employed. A Microsoft Kinect 360 was installed on the head of the PR2, which consists of a linear array of four microphones and three cameras (infrared, depth, RGB). By having a microphone array, the recorded audio has four channels, allowing the possibility of beamforming of the recorded signal. The 600 voice signals were recorded in a room with a volume of 104 m^3^ and a measured reverberation time of 0.5 s, which has the geometry shown in Figure 2. This reverberation time is quite common in indoor environments. We have previously observed, in [75], that training room-independent ASR models by employing RIRs from several indoor environments is feasible. Inside this room, the platform was set up, which is composed of a speech source and a noise source, with a separation of 45° between them from position P1 (two meters from the speech source), which is where the robot was located. Figure 3 shows the actual experimental robotic setup where the database was re-recorded. Two types of scenarios were considered: static, when the robot kept its head fixed, and dynamic, where PR2 was in P1 but rotated its head in a periodical movement. This test bed is a good approximation for a real indoor HRI scenario: in addition to a relative movement between robot head and user, there is reverberation and additive noise. 

### 3.2. Recording Scenarios

Three different scenarios were employed to re-record the telephone dyspnea database with the robotic platform described above. For the static case, two situations were considered (see Figure 2): Static 1, when the robot head was fixed looking at the speech source, i.e., 0°, and Static 2, when the robot head looked at the noise source, i.e., 45°. In the dynamic condition, the robot was also at P1, but its head rotated at constant angular velocity equal to 0.42 rad/s, between −50° and 50° (see Figure 4), so the angles of incidence of the speech and noise waves changed permanently. The static and dynamic conditions are common in real HRI scenarios. The static condition does not need further justification. The angular velocity chosen for the dynamic scenario corresponds to the angular speed of head rotation needed for the robot to follow a virtual target with its head movement. The virtual target would be located two meters away from the robot, and it is moving with a tangential velocity of 3 km/h approximately, which, in turn, is close to walking speed [75,81].

A signal-to-noise ratio (SNR) of 10 dB was adopted for re-recording the database using the robotic platform. To estimate the power of the speech signals, PR2 was placed two meters away from the speech source, i.e., position P1, with its head at 0° (looking straight ahead). Then, the volume of the speech and noise speakers was iteratively adjusted until the desired SNR of 10 dB was obtained. The recorded speech and noise energies were estimated with the 600 vocalizations that were concatenated and 1 minute of restaurant noise from the Aurora database, respectively. 

## 4. System for Respiratory Distress Estimation in HRI Scenarios

Figure 5 illustrates the block diagram of the respiratory distress estimation system in HRI scenarios employed in this study and initially discussed in [77]. Speech enhancement is achieved with a beamforming or spatial filtering scheme that receives three inputs: the target speech source signal; the noise source signal; and the speech source DOA. The enhanced speech signal is fed into the respiratory distress estimation system based on [38] to estimate the user’s mMRC score.

### 4.1. Source Localization and Beamforming

“Jarvis” automatically saves the azimuthal angle of its head, which provides the DOA that is fed into the beamforming or spatial filtering scheme. We assume that source localization is easily performed using the cameras that are commonly available on social robots in conjunction with standard image processing tools. Spatial filtering increases the speech source signal SNR before being input into the respiratory distress estimation system. 

### 4.2. Deep Learning-Based Respiratory Distress Estimation

The system to estimate dyspnea was adapted from the one proposed in [38], which, in turn, is summarized here to describe and explain the differences from the solution adopted in this study. The purpose of the respiratory distress estimation method is to classify users’ dyspnea with the mMRC scale by representing their spontaneous behavior while uttering controlled vocalizations. The user’s’ spontaneous pronunciation behavior appears with the chosen controlled phonetizations and is represented with time-dependent and time-independent speech features. The spontaneous pronunciation behavior includes pauses, variations in intonation and vocalization length, speech rate, and involuntary sounds such as coughing and breathing. The time-dependent features are computed frame-by-frame to capture the dynamics of the vocalization signals and correspond to Mel filters derived from the logarithmic power spectrum of phonetizations /ae-ae/ and /sa-sa/ and the logarithmic power spectrum for the 1-to-30 counting. Time-independent features provide information such as phonation length and intonation variation and slope. Deep learning-based schemes are required to obtain the final mMRC dyspnea score based on the time-dependent and time-independent engineered features that were carefully designed and tested in [38]. In contrast to other approaches in the literature, this method has the advantage that it does not need forced situations or behaviors in an unnatural manner, such as non-spontaneous coughing. Instead, it is based on phonetizations that are simple to reproduce more naturally. The final classifier combines the information provided by time-dependent and time-independent features from the three types of controlled phonetizations. 

The system block diagram is shown in Figure 6. A four-dimensional softmax that represents the probability of each mMRC score is provided by each vocalization type. As mentioned above, the range of the mMRC metric is from zero (healthy) to three (very severe) because the patients that showed mMRC score equal to three and four were merged into a single subset. The following five computations are employed to combine the three phonation-dependent softmax outputs to produce five new softmax values: mean, median, minimum, maximum, and product. The final softmax, whose highest output determines the estimated mMRC score, is generated by averaging these five combinations.

Figure 7 illustrates how the utterance-dependent score is determined. For each phonetization type, there are two classifiers, one for the time-dependent features and another one for the time-independent parameters. Each vocalization type is repeated twice, and their time-dependent and time-independent characteristics are extracted and fed into the corresponding machine learning module to generate one softmax per repetition and feature type. The time-dependent features use a CNN- or LSTM-based architecture, while the time-independent parameters employ an MLP-based scheme. The time-dependent and time-independent softmax outputs for each repetition are separately combined using the same five rules mentioned above to generate a single softmax for each feature type. The vocalization-dependent softmax is computed by averaging the time-dependent and time-independent softmax outputs. This process is repeated for each type of vocalization to obtain the estimated mMRC score. It is worth noting that the training and optimization of the neural-based architectures were carried out with the simulated HRI data described above.

#### 4.2.1. MLP Architectures for Time-Independent Features

The fundamental frequency, F0, which is calculated frame-by-frame with Praat [82], is used to estimate two of the time-independent features according to [38]. The following characteristics are extracted from each vocalization from the F0 curve [37]: the standard deviation and mean of the normalized slope. Then, mean and variance normalization (MVN) was applied to each parameter where the mean and variance of each parameter were computed within the whole database. The phonetization duration in seconds is the third parameter [38]. An MLP was trained here per each type of vocalization, i.e., /ae-ae/, /sa-sa/, and 1-to-30 counting, with the simulated HRI data discussed above. The ADAM optimizer and the cross-entropy loss function were employed. The ReLU activation function was adopted for the hidden layers. Four neurons in the output layer were activated with softmax. The /ae-ae/ MLP had 2 hidden layers with 40 neurons each, and the learning rate was made equal to 0.01. The /sa-sa/ MLP had 2 hidden layers with 20 nodes, and the learning rate was made equal to 0.01. Finally, the 1-to-30 count MLP had 5 hidden layers of 10 nodes each, and the learning rate was made equal to 0.001.

#### 4.2.2. Neural Networks Architectures for Time-Dependent Features

The time-dependent features are based on the FFT log power spectrum and were optimized for each type of phonetization [38]. The 512-sample FFT is estimated in 50 ms windows with 50% overlap where 257 frequency bins are obtained. After that, 14 logarithmic Mel/frame filter energies are calculated for phonetizations /ae-ae/ and /sa-sa/. Mel filters were not used for the one-to-thirty counting vocalization; instead, 75% of the logarithmic spectrum’s lowest frequency bins were chosen, and the corresponding first derivative or delta features were included. The result was 386 features per frame, obtained from 257 bins × 0.75 percent × 2 (second derivate) = 386 features/frame. Parameter means and variances were calculated for the entire database, and MVN was applied to the temporal trajectories of the time-dependent features. Finally, zero padding is performed based on the longest utterance in the training data corresponding to the same type of phonetization. The time-dependent feature architecture and hyperparameter optimization employed the simulated HRI data described above and resulted in the use of neuron stick breaking, cross-entropy as a loss function and ADAM optimizer, and a learning rate of 0.0005 for /ae-ae/ and /sa-sa/ and of 0.0001 for the 1-to-30 counting. Figure 8 depicts the resulting deep learning architectures for the three types of vocalizations.

#### 4.2.3. K-Fold Training with Double Validation

To optimize the available database, a 9-fold cross-validation was performed, from which 11 users were extracted from each of the partitions for testing, except for 1 partition from where 12 individuals were removed. It is important to mention that this data division scheme makes sure that a given speaker could not have vocalizations in the training, validation, or testing subsets simultaneously. In addition to the testing individuals, each partition was composed of training, validation-1, and validation-2 subsets corresponding to 70%, 15%, and 15% of the partition individuals, respectively. The classifiers were trained eight times at each partition to take into consideration the variability due to weight initialization. The training subset was used to estimate the network weights, and validation-1 data were employed to stop the iterations and avoid overfitting with an early stopping of 20 iterations. For each partition, the optimal neural network classifier was chosen among those that resulted from the eight training procedures by picking the one with the highest average accuracy evaluated on the validation-1 and validation-2 subsets. The latter did not make part of the training procedure, so the chosen trained neural network is also the one with the best generalization capability. Then, the test data, which were never seen by the deep learning-based classifier, were propagated to obtain the mMRC scores and metrics for the corresponding partition. These steps were replicated for all the partitions to obtain the scores and metrics for all the 100 individuals. Finally, the whole procedure was repeated five times to obtain more reliable statistics.

#### 4.2.4. Acoustic Modeling Training for Respiratory Distress Estimation in HRI

The telephone database was used to generate simulated training data. The simulation scheme is similar to the one implemented in [75]. The 600 audios of the telephone database were convolved with 33 real impulse responses, which were recorded in static conditions at 1, 2, and 3 meters from the speech source with 11 robot head angles with respect to the speech audio source, which, in turn, results in different DOAs. The head angle was varied from –50° to 50° in steps of 5°. Angle 0° corresponds to the Microsoft Kinect microphones oriented towards the speech source. In addition, additive noise with an SNR within the range of 5 dB and 15 dB was added. This noise is a mixture of the real PR2 noise with different Aurora-4 types of noise (street traffic, train station, car, babble, restaurant, airport). The robot and Aurora noises were summed at an SNR ratio within the range of −5 dB to 5 dB.

#### 4.2.5. Performance Metrics

The metrics adopted to evaluate the system performance were mMRC score accuracy and area under the ROC curve (AUC). Although the classification is carried out with four classes (i.e., mMRC from zero to three), the AUC metric is obtained on a binary basis, where class 0 corresponds to the healthy condition and mMRC from 1 to 3 indicates the presence of dyspnea. Both metrics represent the performance of the respiratory distress estimator. The multiclass accuracy indicates how well the proposed system classifies the severity of dyspnea. On the other hand, the AUC measures how well the system classifies on a binary basis between healthy individuals and dyspnea patients. It is important to highlight that accuracy and AUC are complementary and are not necessarily always correlated. For instance, a system may discriminate more accurately between healthy subjects and dyspnea patients while at the same time not estimating the dyspnea severity particularly well, so it is important to analyze both metrics together. In addition, SNR was used to evaluate the performance of the spatial filtering schemes adopted here. SNR was computed by estimating the noise energy in 0.3 s non-speech intervals at the beginning and end of each phonetization file.

#### 4.2.6. Training and Testing Databases

In this subsection, the training and testing databases and conditions employed here are defined and labeled.

Training database labels: Both telephone and simulated (obtained with the acoustic modeling explained in Section 4.2.4) databases were used to train the respiratory distress neural network-based classifier. The telephone database is denoted as *Telephone_training_data*. The training dataset that resulted from the incorporation of the acoustic model of the HRI scenario (see Section 4.2.4) is named *Simulated_training_data*, which, in turn, corresponds to static simulations. When the D&S or MVDR beamforming scheme responses are also included, i.e., D&S and MVDR, the resulting training database are labeled as *Simulated_training_data + D&S* and *Simulated_training_data + MVDR*, respectively.Testing database labels: The testing telephone database is referred to as *Telephone_testing_data*. For simplicity in presenting the results, the outcomes of experiments using the Static 1 and Static 2 datasets (see Section 3.2) are averaged and presented under the label *static*. Consequently, results with the data re-recorded in real HRI static scenario are referred to as *HRI_static_data*. When the D&S or MVDR beamforming scheme responses are also included, the results are labeled as *HRI_static_data + D&S* and *HRI_static_data + MVDR*, respectively. Similarly, for the dynamic HRI scenario, the corresponding results are *HRI_dynamic_data*, *HRI_dynamic_data + D&S,* and *HRI_dynamic_data + MVDR*. As with the training data, the testing dataset that resulted from the incorporation of the acoustic model of the HRI scenario is named *Simulated_testing_data*, which, in turn, corresponds to static simulations. When the D&S or MVDR beamforming scheme responses are also included, the resulting training database are labeled as *Simulated_training_data + D&S* and *Simulated_training_data + MVDR*, respectively.

## 5. Results and Discussion

The performance of the respiratory distress estimation system in the HRI scenarios studied here are reported and discussed in this section. Each result is achieved by averaging the outcomes obtained by propagating the corresponding testing datasets through the nine optimal deep learning-based classifiers obtained with the nine data partitions. This procedure enables the optimization of the available data and leads to more robust metrics.

### 5.1. Architecture and Hyperparameter Tuning

The optimization of architectures and hyperparameters was performed on a grid search-based analysis for both the time-dependent and time-independent feature-based deep learning classifiers. The best architecture was chosen individually for each classifier by performing training with the database convolved with the RIRs mentioned in Section 4.2.4 but without additive noise. The objective functions were accuracy and AUC obtained with the validation-2 subset, which, in turn, was not seen by the neural network in the training step.

### 5.2. Speech Enhancement with Beamforming Methods

The SNRs estimated on databases *Simulated_testing_data*, *HRI_static_data,* and *HRI_dynamic_data* with and without beamforming schemes (i.e., D&S and MVDR) are shown in Figure 9. As can be seen in Figure 9, the same trend is observed across the three types of datasets. On average, D&S and MVDR provided an increase in SNR equal to 27% and 93%, respectively, when compared to the corresponding databases without beamforming responses incorporated. 

### 5.3. Results with Telephone Training and Real HRI Testing Data

Table 2 shows accuracy and AUC of the respiratory distress estimation system when trained with telephone data and tested on real HRI in static and dynamic conditions with and without the D&S or MVDR beamforming methods, As can be seen in Table 2, the degradation in accuracy and AUC when the telephone testing data are replaced with *HRI_static_data* was equal to 25% and 11%, respectively. The decrease in accuracy and AUC with *HRI_dynamic_data* was equal to 22% and 9%, respectively. The incorporation of D&S beamforming led to an average increase in accuracy and AUC of 9% and 2%, respectively, with both static and dynamic conditions. Similarly, incorporating MVDR beamforming resulted in an average increase in accuracy and AUC of 14% and 2%, respectively, with both static and dynamic conditions.

### 5.4. Results with Simulated Training and Testing Data

Table 3 shows the performance of the respiratory distress estimation system with three matched training/testing conditions, including the beamforming method responses, with *Simulated_training_data*, *Simulated_training_data + D&S*, and, *Simulated_training_data + MVDR*. As expected, these matched experiments show an average improvement in accuracy and AUC when compared with Table 2 where the system was trained and tested with *Telephone_training_data* and *HRI_static_data*, respectively, including the spatial filtering response. The average accuracy and AUC in Table 3 are 4% larger than those in Table 2 when the testing data corresponded to *HRI_static_data*, *HRI_static_data + D&S,* and *HRI_static_data + MVDR*. Surprisingly, in contrast to Figure 9, where MVDR is clearly superior to D&S regarding the SNR gain, results in Table 3 suggest that the difference between both spatial filtering schemes is less substantial with respect to the respiratory distress evaluation performance. This is likely to be a consequence of the fact that the experiments in Table 3 were carried out with training/testing matched conditions, including the beamforming scheme. 

### 5.5. Results with Simulated Training and Testing with Real Data

Table 4 shows the performance of the system when trained with simulated data and tested with real static HRI conditions with and without the beamforming scheme responses. As can be seen in Table 4, the average accuracy and AUC are almost the same as those in Table 3. When compared with Table 2, where the training data corresponded to the telephone database, the increase in average accuracy and AUC with the real static HRI dataset was 6% and 5%, respectively. Particularly, the comparison with *Telephone_training_data* (Table 2) shows that *Simulated_training_data* (Table 4) led to an increase of 11% and 5% in accuracy and AUC, respectively with the static HRI testing dataset. These results confirm the pertinence of the acoustic modeling incorporated into the training data to approximate the real HRI scenario in static conditions from the respiration assessment point of view. This is especially interesting because it is difficult to recruit and record speech from patients with dyspnea even more in HRI environments. Beamforming methods D&S and MVDR led to an improvement in accuracy of 7% and 10%, respectively, and just a small increase in AUC of 1% and 3%, respectively. As in Table 3, the difference between D&S and MVDR with respect to accuracy and AUC is small when they are applied in both training and testing. 

Table 5 shows the performance of the system when trained with simulated data and tested with real dynamic HRI conditions. The results in Table 5 are very similar to those obtained with the static HRI scenario (Table 4). It is possible to observe that the use of beamforming schemes leads to similar improvements in both the static and dynamic HRI conditions. Unexpectedly, and in contrast to what was observed with ASR in a similar HRI scenario [75,81], the experiments with the dynamic HRI dataset did not demonstrate worse accuracy and AUC than those obtained with the static HRI dataset. The respiratory distress estimation technology evaluated here showed a baseline degradation and improvements due to the incorporation of acoustic modeling in the training data that are quite similar for the static (Table 4) and dynamic (Table 5) scenarios. In ASR, the search is performed on a frame-by-frame basis. Consequently, if the acoustic conditions are time-dependent, the ASR search in those frames with lower local SNR or higher reverberation effect may condition the search in the following frames until the end of the utterance. Nevertheless, the deep learning-based classifiers deliver the estimated mMRC metric on an utterance-by-utterance basis. Despite the fact that the classifiers employ time-independent and time-dependent features computed over short-term windows, the period of the robot head rotational movements is approximately equal to seven seconds. This suggests that more than one complete back and forth robot head movement takes place during a controlled vocalization (i.e., 12.7 s on average). As a result, there could be an acoustic severity compensation effect along the whole utterances to estimate respiratory distress.

As can be seen in Table 3, Table 4 and Table 5, incorporating the target environment acoustic model in the training utterances is as helpful as the use of the beamforming schemes. On the other hand, there are many noise or channel robust methods for single-channel applications such as those found in telephony. However, the improvement provided by these techniques is generally quite limited when they are applied in both training and testing, as done here [83,84]. In this context, evaluating other noise removal methods for single channels is out of the scope of the current study. 

As discussed in [38], the time-dependent and time-independent features provide an important complementarity, and the combination of both types of parameters led to better results in accuracy and binary classification. The analysis in the real HRI scenarios considered here is more complex. As shown in Figure 10, the combination of classifiers based on time-dependent and time-independent features did not lead to better accuracy than was obtained using each classifier individually. In fact, the best result was obtained using time-dependent features only. Regarding the binary dyspnea presence–absence classification, the results of Figure 11 show that the use of time-independent features alone and in combination with time-dependent parameters led to the same AUC score, which, in turn, is greater than the one obtained with time-dependent features only. This result is due to the fact that the time-dependent parameter classifier appears to prioritize the class mMRC equal to zero, leading to a worse AUC curve. However, the best joint accuracy and AUC metrics are obtained using the combination of both types of classifiers. 

## 6. Conclusions

In this paper, the respiratory distress estimation methodology previously proposed in [38] is adapted and evaluated in real static and dynamic HRI scenarios. First, the architecture and hyperparameters of the deep learning-based classifiers were tuned using the original dyspnea telephone data that had incorporated an acoustic model of the target indoor room. Experiments with the telephone dataset re-recorded using our robotic platform show that training the respiratory distress classifiers with the original telephone data plus the target acoustic environment model leads to average accuracy and AUC that are just 0.4% smaller than those obtained with matched training/testing conditions with simulated data. These results basically corroborate the suitability of the acoustic modeling incorporated into the training data discussed here to approximate the real HRI scenarios from the respiration difficulty assessment point of view. We also found that there was not much difference in accuracy and AUC between the static and dynamic HRI conditions. In addition, the delay-and-sum and MVDR beamforming methods lead to average improvements in accuracy and AUC equal to 8% and 2%, respectively, when applied to both training and testing data. Finally, for most of the experiments, a combination of classifiers based on time-independent and time-dependent features provided the best joint accuracy and AUC score. The evaluation of other speech enhancement methods for distant speech processing and the determination of the causes of respiratory distress are potential topics for future research.

## Figures and Tables

**Figure 1 sensors-23-07590-f001:**
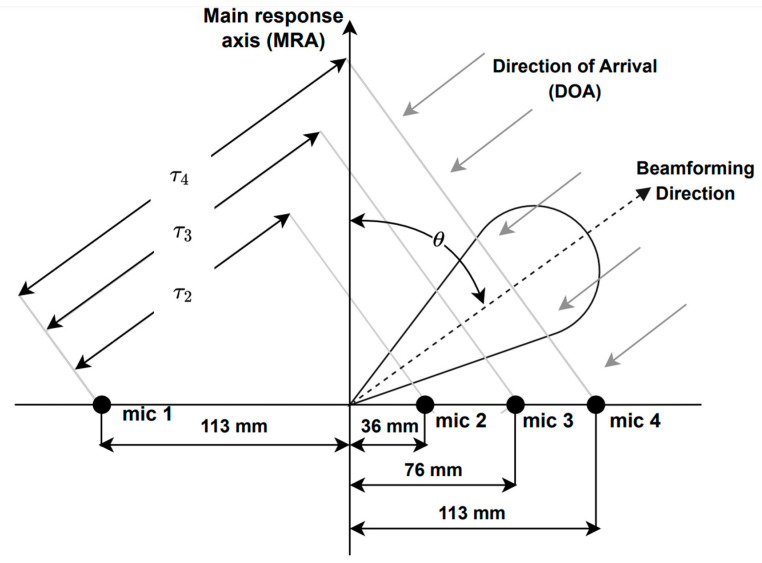
Microphone array geometry of the Microsoft Kinect, where $\tau_{l}$  is the time delay between microphone l and the reference one, i.e., microphone 1, and ϕ is the look direction or DOA.

**Figure 2 sensors-23-07590-f002:**
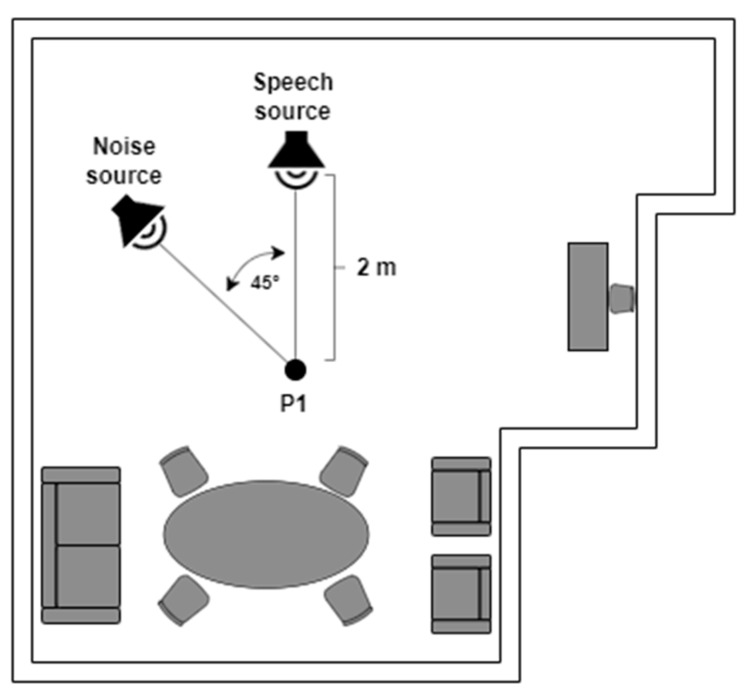
Indoor HRI scenario.

**Figure 3 sensors-23-07590-f003:**
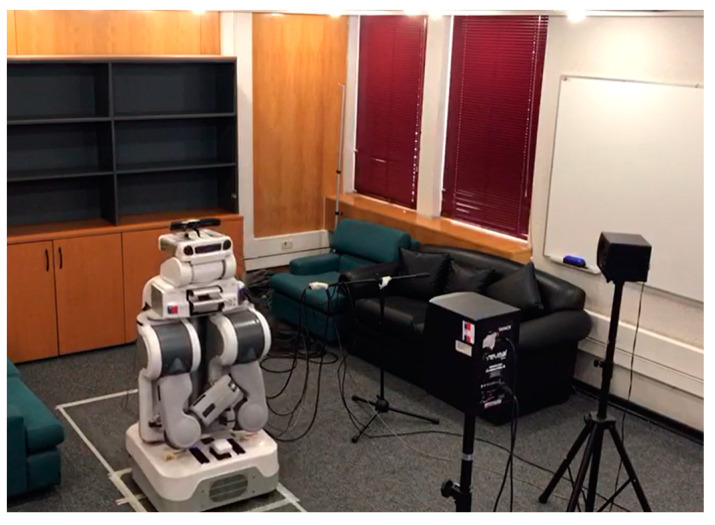
The experimental robotic setup employed to re-record the original telephone database.

**Figure 4 sensors-23-07590-f004:**
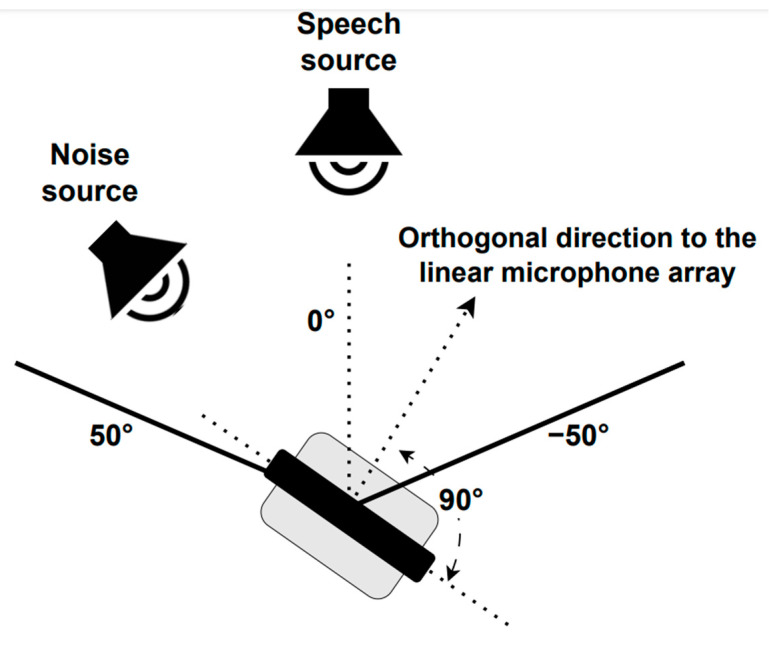
PR2 robot head movement.

**Figure 5 sensors-23-07590-f005:**
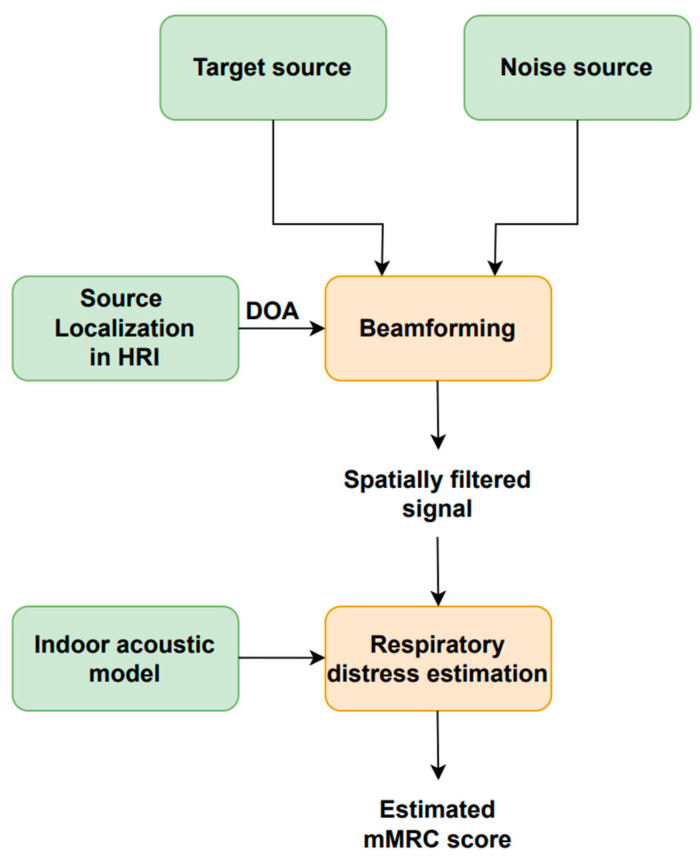
Block diagram of the respiratory distress estimation system for HRI scenarios.

**Figure 6 sensors-23-07590-f006:**
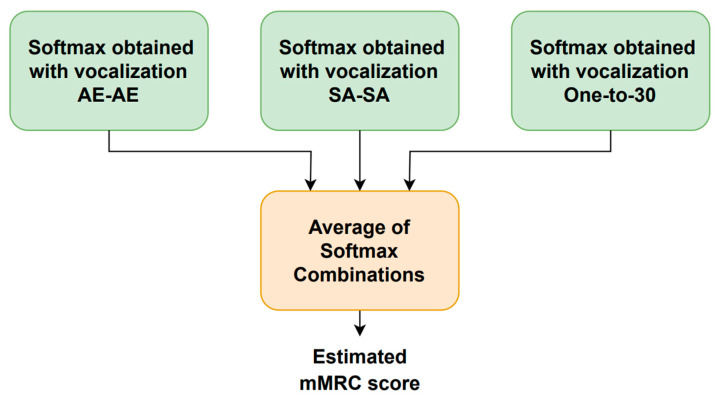
Block diagram of the respiratory distress estimation system.

**Figure 7 sensors-23-07590-f007:**
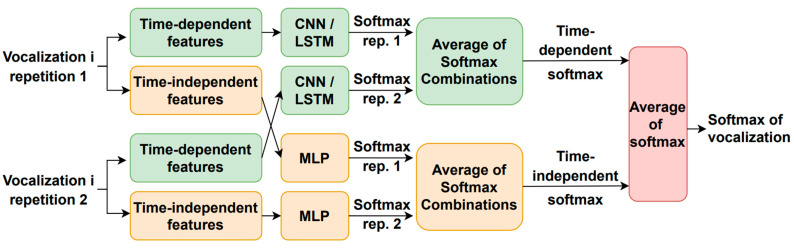
Estimation of the vocalization-dependent scores.

**Figure 8 sensors-23-07590-f008:**
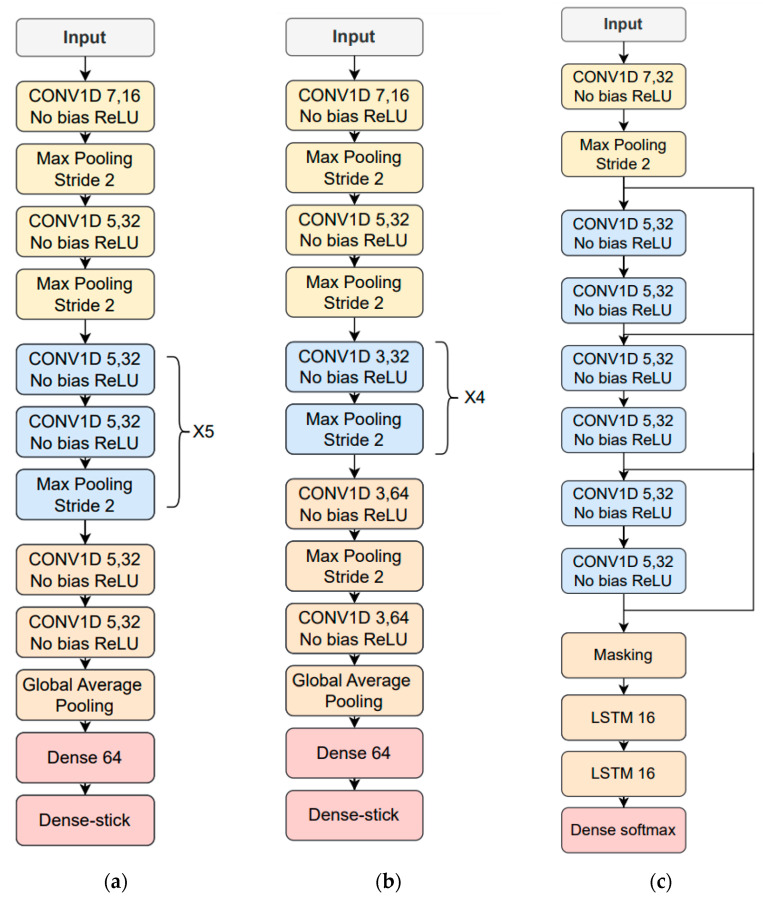
Resulting neural network architectures after being tuned with the simulated data: (**a**) /ae-ae/ phonetization; (**b**) /sa-sa/ phonetization; (**c**) 1-to-30 counting vocalization.

**Figure 9 sensors-23-07590-f009:**
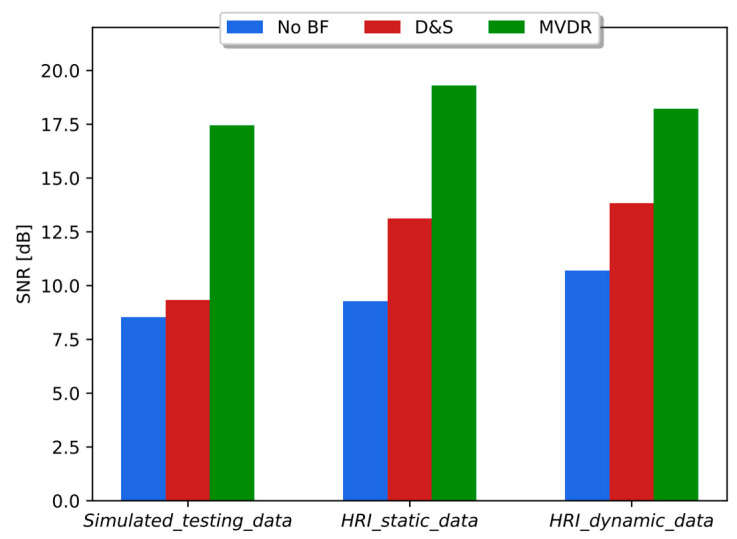
SNR obtained without (no BF) and with beamforming methods (D&S and MVDR) with simulated data and HRI datasets, i.e., static and dynamic scenarios.

**Figure 10 sensors-23-07590-f010:**
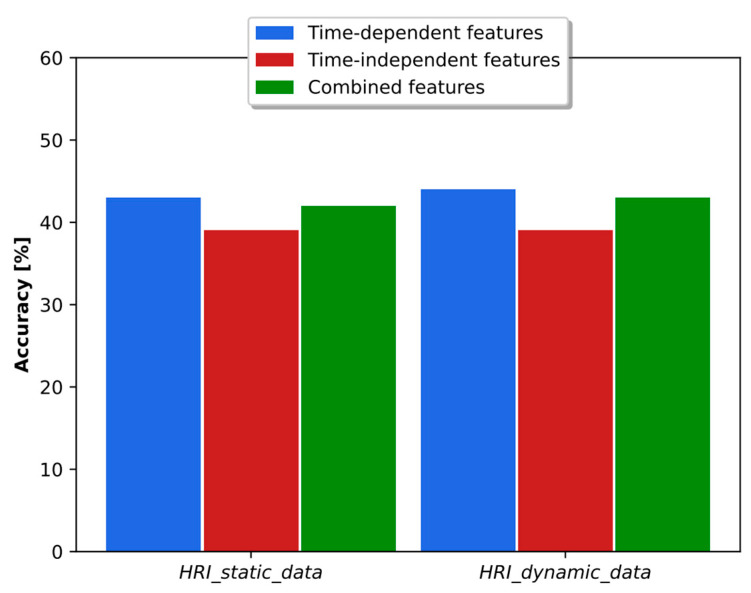
Accuracy obtained with time-dependent- and time-independent-based classifiers individually and in combination when the respiratory distress estimation system was trained with simulated data and tested with the real static and dynamic HRI scenarios.

**Figure 11 sensors-23-07590-f011:**
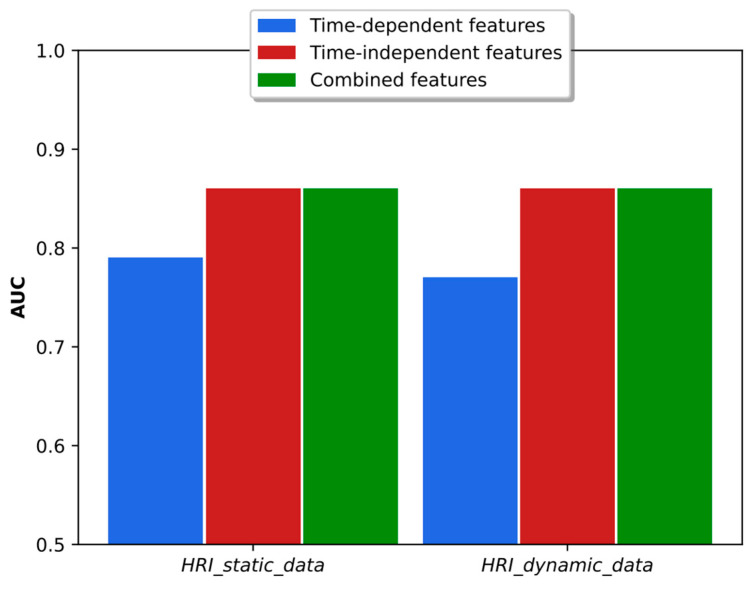
AUC obtained with time-dependent- and time-independent-based classifiers individually and in combination when the respiratory distress estimation system was trained with simulated data and tested with the real static and dynamic HRI scenarios.

**Table 1 sensors-23-07590-t001:** Demographic and clinical info of the patients. FEV1 = forced expiratory volume in 1 s and FVC = forced vital capacity (spirometric values were available for 59 patients).

Diagnosis	n	Age, Years Average SD	Females (n)	Smoking (n)	Pack Year Index (Average ± SD)	FEV_1_/FVC (Average ± SD)	FEV_1_ % Pred (Average ± SD)	FVC % Pred (Average ± SD)
COPD	43	74.8 ± 8.1	15	43	34.11 ± 21.8	54.3 ± 11.1	65.1 ± 22.9	94.7 ±21.8
PF	19	64.6 ± 8.6	9	9	17.8 ± 14.7	88.7 ± 4.3	88.9 ± 22.7	78.9 ± 20
COVID-19	4	56.25 ± 8.9	3	2	20 ± 10	88.5 ± 2.1	95.5 ± 17.7	89.5 ± 14.8
Total	66	70.7 ± 10	27	54	31.4 ± 21.4	63.8 ± 18.3	71.1 ± 24.8	90.4 ± 22

**Table 2 sensors-23-07590-t002:** Accuracy and AUC when the respiratory difficulty estimation system was trained with the original telephone data and tested in real HRI scenarios.

Training Data	Testing Data	Accuracy (%)	AUC
*Telephone_training_data*	*Telephone_testing_data*	51	0.92
*Telephone_training_data*	*HRI_static_data*	38	0.82
*Telephone_training_data*	*HRI_static_data+D&S*	42	0.84
*Telephone_training_data*	*HRI_static_data+MVDR*	47	0.86
*Telephone_training_data*	*HRI_dynamic_data*	40	0.84
*Telephone_training_data*	*HRI_dynamic_data+D&S*	43	0.85
*Telephone_training_data*	*HRI_dynamic_data+MVDR*	42	0.84

**Table 3 sensors-23-07590-t003:** Accuracy and AUC when the respiratory distress estimation system was trained and tested with simulated data.

Training Data	Testing Data	Accuracy (%)	AUC
*Simulated_training_data*	*Simulated_testing_data*	41	0.86
*Simulated_training_data+D&S*	*Simulated_testing_data+D&S*	46	0.87
*Simulated_training_data+MVDR*	*Simulated_testing_data+MVDR*	45	0.91

**Table 4 sensors-23-07590-t004:** Accuracy and AUC when the respiratory distress estimation system was trained with simulated data and tested with the real static HRI scenario.

Training Data	Testing Data	Accuracy (%)	AUC
*Simulated_training_data*	*HRI_static_data*	42	0.86
*Simulated_training_data+D&S*	*HRI_static_data+D&S*	45	0.87
*Simulated_training_data+MVDR*	*HRI_static_data+MVDR*	46	0.89

**Table 5 sensors-23-07590-t005:** Accuracy and AUC when the respiratory distress estimation system was trained with simulated data and tested with the real static HRI scenario.

Training Data	Testing Data	Accuracy (%)	AUC
*Simulated_training_data*	*HRI_dynamic_data*	43	0.86
*Simulated_training_data+D&S*	*HRI_dynamic_data+D&S*	44	0.87
*Simulated_training_data+MVDR*	*HRI_dynamic_data+MVDR*	44	0.87

## Data Availability

Not applicable.

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
