# Peer review of "Automatic Detection of Dyspnea in Real Human–Robot Interaction Scenarios"

_sensors, 2023, doi:10.3390/s23177590_

Round 1

Reviewer 1 Report

This paper aims to propose a deep-learning-based dyspnea detection algorithm to, first, detect the existence of respiratory diseases such as COPD, pulmonary fibrosis, COVID-19, etc., through asking the patient to pronounce key phonetizations that may involuntarily provoke symptoms in case there is dyspnea, and second, assign a mMRC dyspnea severity level to the patient. A simulated HRI dataset was created by convolution to train the classification network, and a real HRI scenario dataset was created by re-recording previous telephone datasets in a room with environmental noise to test the network. The use of microphone array hardware and beamforming speech enhancement methods to preprocess the training and testing dataset were experimented with and found to yield favorable outcomes. The dataset generation methodology and classification models are presented, and results discussed, but several concerns remain to be addressed.

Major concerns

- Statements explaining the necessity of this research in the real world is not fully convincing. For example, only patients with slight symptoms may need to know if they have caught respiration diseases; patients with serious condition (i.e., at mMRC levels above 2) may need to know the disease exactly to find cure options, or attend hospital to seek treatment, instead of knowing the severity level.

- Dataset collection room setup seems rather randomly decided (figures 2 - 4), causing ambiguity in whether the test set generated can represent “real HRI scenarios”. Parameters in both static and dynamic scenarios need more reasoning, or more experiment cases with different speech-noise-robot setup should be used for confirmation.

- More attention should be given to speech enhancement and other differences in “real HRI scenarios”, in the review and other sections of this paper, since this is the main topic of this paper and the incremental work on top of your previous paper Dyspnea Severity Assessment Based on Vocalization Behavior with Deep Learning on the Telephone.

Good writting.

Author Response

Automatic detection of dyspnea in real human-robot interaction scenarios

MS Number: 2518125

Eduardo Alvarado, Nicolás Grágeda, Alejandro Luzanto, Rodrigo Mahu, Jorge Wuth, Laura Mendoza, Richard Stern and Néstor Becerra Yoma

REPLY TO COMMENTS

We thank the editor for handling the review process and the anonymous reviewers for their valuable feedback on our manuscript. We have carefully considered all the comments and provide below our point-by-point response to each suggestion or observation. Furthermore, we incorporated almost all the suggestions in the revised version of the manuscript. The text that was included is in red font to facilitate the review of the revised manuscript, and to emphasize our willingness and effort to address all the reviewers´ comments.  Finally, the English writing was proofread carefully.

COMMENTS FOR THE AUTHOR:

REVIEWER #1

“This paper aims to propose a deep-learning-based dyspnea detection algorithm to, first, detect the existence of respiratory diseases such as COPD, pulmonary fibrosis, COVID-19, etc., through asking the patient to pronounce key phonetizations that may involuntarily provoke symptoms in case there is dyspnea, and second, assign a mMRC dyspnea severity level to the patient. A simulated HRI dataset was created by convolution to train the classification network, and a real HRI scenario dataset was created by re-recording previous telephone datasets in a room with environmental noise to test the network. The use of microphone array hardware and beamforming speech enhancement methods to preprocess the training and testing dataset were experimented with and found to yield favorable outcomes. The dataset generation methodology and classification models are presented, and results discussed, but several concerns remain to be addressed.”

“Major concerns”

“Statements explaining the necessity of this research in the real world is not fully convincing. For example, only patients with slight symptoms may need to know if they have caught respiration diseases; patients with serious condition (i.e., at mMRC levels above 2) may need to know the disease exactly to find cure options, or attend hospital to seek treatment, instead of knowing the severity level.”

The fact that the detection is carried out by means of audio analysis makes it possible to standardize the evaluation, reducing variability or bias between different doctors who take a test like a questionnaire. While the dyspnea assessment questionnaire is easy for a doctor to apply, it is not particularly easy for ordinary people to use, as the civilian population may have difficulties in understanding or answering the questions, particularly in the case of elderly adults. Moreover, the response given in a questionnaire can be influenced by the patient's mood or habituation to the disease [1-3].  There already a number of studies that focus on remote monitoring of respiratory diseases by means of audio analysis [4], which indicates that this is a reasonable and important topic for current research. The early identification of dyspnea can be helpful to identify diseases that cause it (e.g.  respiratory, cardiac issues, etc.) and to allow their timely management. Moreover, monitoring dyspnea is very important to detect acute diseases (for example, pneumonia due to COVID-19 and other pneumonias) and thus to determine if it is necessary to hospitalize the patient. It can also be useful to follow-up the patients´ evolution and to manage chronic conditions. In the case of HRI, a clear application could be in a health center where a robot can interview patients without the need for a health professional. As the reviewer indicates, an increase in the measured mMRC score can trigger immediate actions to treat the patients. However, it is worth highlighting that the applicability of user profiling goes beyond the health sector. Consider, for instance, robots and human beings working together in a common task in a commercial or military application. Robots may employ the estimated respiratory distress of the persons that they interact with to optimize, for example, the task assignments. We note that physical tiredness also leads to some degree of respiratory difficulties. While prompting the users to pronounce controlled vocalizations may not be feasible in some scenarios, this research is an important step toward the respiratory distress evaluation in human-robot collaboration contexts. Part of this discussion was incorporated into the revised version of the manuscript to clarify this issue.

“Dataset collection room setup seems rather randomly decided (figures 2 - 4), causing ambiguity in whether the test set generated can represent “real HRI scenarios”. Parameters in both static and dynamic scenarios need more reasoning, or more experiment cases with different speech-noise-robot setup should be used for confirmation”

Thank you for this comment. The conditions of data collection and analysis were not arbitrarily selected and are part of a series of studies spanning several years. The data set collection set up has two aspects: the room and the robot. The room is the same one that had been employed in other studies [5-7] and provides a reverberation time equal to 0.5 seconds, and was selected because it is typical for relevant indoor environments. A room needed to be chosen for this research and this one seemed appropriate. It is also worth noting that results from a previous study [5], demonstrated that useful room-independent models can be trained by employing room impulse responses (RIR) from several indoor environments. The robot itself is a sophisticated and widely used commercial experimental platform whose operating conditions need to be understood for deployment in practical applications. The static and dynamic conditions are common in real HRI scenarios. The static condition does not need further justification. The angular velocity chosen for the dynamic scenario corresponds to the angular speed of head rotation needed for the robot to follow a virtual target with its head movement. The virtual target would be located two meters away from the robot, moving with tangential velocity of 3 km/h approximately, which in turn is close to the walking speed [5]. At this point we need to emphasize that many experimental issues needed to be addressed to obtain the results provided in the current manuscript. We believe that additional operating conditions with small differences among them will not provide useful insights beyond those that we outline in the present paper. However, observe that more diverse robot and noise conditions were employed in the simulated training and testing setting. Part of this discussion was included in the revised version of the manuscript to clarify this issue.

“More attention should be given to speech enhancement and other differences in “real HRI scenarios”, in the review and other sections of this paper, since this is the main topic of this paper and the incremental work on top of your previous paper Dyspnea Severity Assessment Based on Vocalization Behavior with Deep Learning on the Telephone

Thank you for the opportunity to comment on this issue. Static and dynamic human-robot interaction scenarios are completely different from telephony and impose important challenges such as distant speech processing and time-varying acoustic channels (TVAC) [5-6]. Distant speech processing was addressed here (at least in part) through the use of the D&S and MVDR with beamforming schemes.  We assume that source localization is easily performed using the cameras that are commonly available on social robots in conjunction with standard image processing tools, as noted in the paper.  Beamforming schemes increase the speech source signal SNR. To make this issue clearer, Figure 5 in the modified paper was modified by changing “Enhanced signal” with “Spatially filtered signal”. To address the TVAC problem, the incorporation of room impulse responses (RIRs) and additive noise was evaluated in the training procedure. The RIRs employed here were measured at different robot head angles and robot-source distances [5-6]. As can be seen in the current manuscript, incorporating the target environment acoustic model in the training utterances is as relevant as the beamforming schemes. On the other hand, the reviewer is right in the sense that there are many noise or channel robust methods for single channel applications such as those found in telephony. However, we have observed that the improvement obtained by this kind of techniques is quite limited when they are applied to both training and testing data as is done here [8-9]. In this context, evaluating noise-removal methods for single channels is proposed for future research. This discussion was included in the revised version of the manuscript.

REFERENCES

[1]        Stoeckel M. C., Esser R. W., Gamer M., Büchel C., and von Leupoldt, A. Brain mechanisms of

short-term habituation and sensitization toward dyspnea. Frontiers in psychology, 2015, 6, 748. https://doi.org/10.3389/fpsyg.2015.00748

[2]         Wan  L., Stans  L., Bogaerts  K., Decramer  M., and Van den Bergh  O. Sensitization in medically      unexplained dyspnea: differential effects on intensity and unpleasantness. Chest, 2012, 141(4), 989–995. https://doi.org/10.1378/chest.11-1423

[3]         Von Leupoldt  A., Dahme  B. Psychological aspects in the perception of dyspnea in obstructive pulmonary diseases. Respir Med. 2007, 101(3), 411-22. doi: 10.1016/j.rmed.2006.06.011. Epub 2006 Aug 8. PMID: 16899357.

[4]           Xia T., Han J., Mascolo C. Exploring machine learning for audio-based respiratory condition screening: A concise review of databases, methods, and open issues. Exp Biol Med (Maywood). 2022, 247(22), 2053-2061.

[5]           Novoa, J.; Mahu, R.; Wuth, J.; Escudero, J.P.; Fredes, J.; Yoma, N.B. Automatic Speech Recognition for Indoor Hri Scenarios. ACM Transactions on Human-Robot Interaction (THRI) 2021, 10, 1–30.

[6]           Novoa, J.; Wuth, J.; Escudero, J.P.; Fredes, J.; Mahu, R.; Yoma, N.B. DNN-HMM Based Automatic Speech Recognition for HRI Scenarios. In Proceedings of the ACM/IEEE International Conference on Human-Robot Interaction; 2018.

[7]           Díaz, A.; Mahu, R.; Novoa, J.; Wuth, J.; Datta, J.; Yoma, N.B. Assessing the Effect of Visual Servoing on the Performance of Linear Microphone Arrays in Moving Human-Robot Interaction Scenarios. Comput Speech Lang 2021, 65, 101136.

[8]           Fredes, J., Novoa, J., King, S., Stern, R.M.,  Yoma, N.B. "Locally Normalized Filter Banks Applied to Deep Neural-Network-Based Robust Speech Recognition," IEEE Signal Processing Let-ters, vol. 24, no. 4, pp. 377-381, 2017.

[9]           Novoa, J., Fredes, J., Poblete, V., & Yoma, N. B. “Uncertainty weighting and propagation in DNN-HMM-based speech recogni-tion”. Computer Speech & Language, Volume 47, January 2018.

Reviewer 2 Report

In this report, I would like to note some critical facts regarding the review of the article "Automatic detection of dyspnea in real human-robot interaction scenarios" submitted to the journal Sensors.

It is worth noting that the theoretical background of the subject that this article deals with is very well structured, comprehensively, and stepwise presented by the authors. The reader is led naturally to the subject, introducing him to information and knowledge with documented and modern bibliography. The study tries to cover all sides and aspects of the subject critically and in an organized manner.

The literature review and presentation of existing knowledge around the topic is clear and instructive, which, combined with the organization of topics and paragraphs that have been chosen structurally, helps the reader a lot. It is also essential that the text follows the journal standards and that the writing language is of a very good standard.

The authors clearly connect the theoretical framework with the concerns and gaps in the literature while they have also distinguished articles that have strong extrapolated results to inform the reader, which gives more value to the present study.

The resulting research questions are carefully recorded and distinct from each other.

In the experiment presented, it is very important that the research team has a clear direction and reference to the literature and methodology on which it was based.

It would be essential to improve the presentation of the experiment and its content to make a brief description and presentation of the 200 audio materials used or later at a second audio level (600).

It would be good to have a more homogeneous number in the sample and to have a power study done in relation to the sampling, especially since the number of people with covid19 was very small. Thus, the results of the research would have more power and credibility.

It would also be necessary for the research team to present a few primary demographic data about the sample that are missing, such as the age, gender, and cognitive background of the patients as well as their educational background, something that is usually used as a factor of exclusion in similar studies.

Indeed, these data and the corresponding comparison based on these factors would strengthen the conclusions of the study.

The study results are presented with simplicity, are comprehensive and scalable, and help the reader understand better. Their manner and structure are adequate and transparent.

The study conclusions and implications for future research are satisfactory and descriptive with precise analysis. If I may, I would suggest that the phrase “Quite surprisingly” in lane 657 can be removed as it does not match the style and weight of the extracts from your study.

Overall, a very good and structured study that you should be proud of.

Author Response

Automatic detection of dyspnea in real human-robot interaction scenarios

 MS Number: 2518125

Eduardo Alvarado, Nicolás Grágeda, Alejandro Luzanto, Rodrigo Mahu, Jorge Wuth, Laura Mendoza, Richard Stern and Néstor Becerra Yoma

REPLY TO COMMENTS

We thank the editor for handling the review process and the anonymous reviewers for their valuable feedback on our manuscript. We have carefully considered all the comments and provide below our point-by-point response to each suggestion or observation. Furthermore, we incorporated almost all the suggestions in the revised version of the manuscript. The text that was included is in red font to facilitate the review of the revised manuscript, and to emphasize our willingness and effort to address all the reviewers´ comments.  Finally, the English writing was proofread carefully.

COMMENTS FOR THE AUTHOR:

REVIEWER #2

“In this report, I would like to note some critical facts regarding the review of the article "Automatic detection of dyspnea in real human-robot interaction scenarios" submitted to the journal Sensors.”

“It is worth noting that the theoretical background of the subject that this article deals with is very well structured, comprehensively, and stepwise presented by the authors. The reader is led naturally to the subject, introducing him to information and knowledge with documented and modern bibliography. The study tries to cover all sides and aspects of the subject critically and in an organized manner.”

“The literature review and presentation of existing knowledge around the topic is clear and instructive, which, combined with the organization of topics and paragraphs that have been chosen structurally, helps the reader a lot. It is also essential that the text follows the journal standards and that the writing language is of a very good standard.”

“The authors clearly connect the theoretical framework with the concerns and gaps in the literature while they have also distinguished articles that have strong extrapolated results to inform the reader, which gives more value to the present study.”

“The resulting research questions are carefully recorded and distinct from each other.”

“In the experiment presented, it is very important that the research team has a clear direction and reference to the literature and methodology on which it was based.”

Thank you for your comments.

“It would be essential to improve the presentation of the experiment and its content to make a brief description and presentation of the 200 audio materials used or later at a second audio level (600).”

Thank you for your comment, but we are not quite sure how we can improve the presentation of the experiments and databases considering that we followed the same strategy as in our previous paper [10]. This is why we refer to it: to avoid replicating content.

“It would be good to have a more homogeneous number in the sample and to have a power study done in relation to the sampling, especially since the number of people with covid19 was very small. Thus, the results of the research would have more power and credibility.”

It is also possible that the manifestation of dyspnea does not depend on the illness that causes it. Discriminating dyspnea depending on its cause is out of the scope of this paper. However, this hypothesis still seems reasonable. The aim of the technology presented here was to detect dyspnea in HRI scenarios independently of the underlying cause. Nevertheless, the determination of possible differences in observed dyspnea with respect to gender, age, comorbidities, etc, can be considered as a topic for future research. This discussion was incorporated in the current version of the manuscript. We need to emphasize that obtaining this kind of clinical data is really difficult and time consuming!  It also requires an infrastructure that is not easy to achieve. As in with every study involving actual medical patients, compromises must be made because the specifics of the limited number of patients that are actually available is typical out of our control.  Actually, it can be observed that many clinical studies employ tens of patients and no more than that for these very reasons.

“It would also be necessary for the research team to present a few primary demographic data about the sample that are missing, such as the age, gender, and cognitive background of the patients as well as their educational background, something that is usually used as a factor of exclusion in similar studies. Indeed, these data and the corresponding comparison based on these factors would strengthen the conclusions of the study.”

Thank you for the comment. A table with the patients´ demographic and clinical information was added to the revised version of the manuscript.

“The study results are presented with simplicity, are comprehensive and scalable, and help the reader understand better. Their manner and structure are adequate and transparent.”

“The study conclusions and implications for future research are satisfactory and descriptive with precise analysis. If I may, I would suggest that the phrase ‘Quite surprisingly’ in lane 657 can be removed as it does not match the style and weight of the extracts from your study.”

We incorporated your suggestion. Thank you!

REFERENCES

[10]      Alvarado, E.; Grágeda, N.; Luzanto, A.; Mahu, R.; Wuth, J.; Mendoza, L.; Yoma, N.B. Dyspnea Severity As-sessment Based on Vocalization Behavior with Deep Learning on the Telephone. Sensors 2023, 23, doi:10.3390/s23052441.

Round 2

Reviewer 1 Report

The authors repaired the paper very carefully.
So I think this paper could be accepted for this journal.
I have no further comments to this paper.

Good